# Social isolation and cardiometabolic burden synergistically predict physical dysfunction in aging Chinese adults: Evidence of risk thresholds and the mediating role of frailty

Shuang Deng[1], Zhongqiang Guo[2]*, Siyuan He[2], Xuetao Sun[3]

**1** Tianhua College, Shanghai Normal University, Shanghai, China, **2** School of Nursing and Health, Henan University, Kaifeng, Henan, China, **3** The 908th Hospital of the Joint Logistics Support Force, Nanchang, Jiangxi, China

* guozhongqiang0701@gmail.com

## Abstract

### Objective

This study aimed to investigate the independent and synergistic effects of social isolation and multidimensional biomarkers (cardiovascular, metabolic, renal, muscular, and frailty) on physical dysfunction in middle-aged and older Chinese adults by utilizing an integrated sociobiological framework to address the limitations of the current research.

### Method

A cross-sectional analysis was conducted using nationally representative data from the China Health and Retirement Longitudinal Study (CHARLS 2015; N = 3,756 participants aged ≥45 years). Physical dysfunction was defined as difficulty in ≥1 of 9 basic activities of daily living. Core exposures included social isolation (composite score), cardiovascular–kidney–metabolic (CKM) syndrome stage (0–4), vascular ageing (estimated pulse wave velocity [EPWV]), renal function (eGFR), body composition (appendicular skeletal muscle mass [ASM]), metabolic status (visceral adiposity index [VAI] and C-reactive protein triglyceride glucose index [CTI]), and frailty (frailty index). Multivariable logistic regression adjusted for demographic, lifestyle, and socioeconomic factors. Threshold effect models revealed nonlinear relationships. Causal mediation analysis (1000 bootstraps) was used to quantify pathway effects.

### Results

Social isolation independently increased physical dysfunction risk by 38% (adjusted odds ratio [aOR]=1.380; 95% CI: 1.132–1.683; *P* = 0.002), with stronger effects in those aged <60 years (OR=1.731), males (OR=1.400), and rural residents

**Data availability statement:** The minimal data set required to replicate all study findings, including the underlying values for all statistical analyses, figures, and tables, has been deposited to the Figshare repository. The data are now publicly available at the following URLs and DOIs: Raw Data: https://doi.org/10.6084/m9.figshare.30273184.v1 Fig 1. Flow Diagram for Participant Inclusion: https://doi.org/10.6084/m9.figshare.30273214.v1 Fig 2. Smooth Curve Fitting: https://doi.org/10.6084/m9.figshare.30273235.v1 Fig 3. Subgroup Analysis: https://doi.org/10.6084/m9.figshare.30273238.v1 Fig 4. Causal Mediation Analysis: https://doi.org/10.6084/m9.figshare.30273253.v1 Table 1. Baseline Characteristics: https://doi.org/10.6084/m9.figshare.30273304.v1 Table 2. Multiple Logistic Regression Equation: https://doi.org/10.6084/m9.figshare.30273310.v1 Table 3. Threshold Effect Analysis: https://doi.org/10.6084/m9.figshare.30273313.v1 Supplementary Table 1. Sensitivity Analysis: https://doi.org/10.6084/m9.figshare.30273316.v1 README File (providing context and descriptions for the datasets): https://doi.org/10.6084/m9.figshare.30273265.v1.

**Funding:** The author(s) received no specific funding for this work.

**Competing interests:** The authors have declared that no competing interests exist.

(OR=1.679). Advanced CKM stage 4 was associated with a 4.8-fold increased risk (aOR=4.805, 95% CI: 2.691–8.579; $P < 0.001$). Key biomarker thresholds were identified: EPWV had an inflection point at 7.178 m/s, with 102.6% increased risk per unit below this threshold (OR=2.026; $P = 0.021$). A frailty index of <7.679 increased risk by 112.4% per unit (OR=2.124; $P < 0.001$). Frailty mediated 57.8% ($\beta = 0.052$, $P < 0.001$) of the effect of EPWV on dysfunction. ASM loss beyond 22.94 kg increased risk (OR=1.166, $P = 0.008$).Sensitivity analyses using E-values indicated that unmeasured confounding was unlikely to fully explain the observed associations.

## Conclusion

Social isolation and multidimensional biomarkers (particularly CKM severity, vascular stiffness, and frailty) synergistically drive physical dysfunction in ageing Chinese adults. Frailty is a critical mediator of the impact of vascular dysfunction. The identified biomarker thresholds (e.g., EPWV = 7.178 m/s) offer intervention windows. Integrated strategies combining social connections (e.g., community support) with biomarker screening and targeted interventions (e.g., anti-frailty training for elevated EPWV) are essential to disrupt the "isolation–comorbidity–dysfunction" cycle.

## Introduction

The accelerated ageing of the global population has emerged as among the most remarkable social changes of the 21st century. According to the United Nations World Population Prospects report, the proportion of people aged 60 and above increased from 9.2% in 1990 to 11.7% in 2013 and is expected to reach 21.1% by 2050. At that time, the number of elderly people will have increased from 841 million in 2013 to more than 2 billion, surpassing the number of children for the first time [1,2]. This trend is particularly prominent in developing countries, where two-thirds of the elderly population currently resides, and this proportion is expected to increase to nearly 80% by 2050. As one of the countries with the fastest ageing process, China's population aged 65 and above is expected to exceed 30% by 2035, highlighting the severity of these trends. The core public health challenge accompanying ageing is the prevalence of physical dysfunction, characterized by progressive loss of daily activity abilities such as walking, dressing, and eating. According to data from the China Health and Retirement Longitudinal Study (CHARLS), the prevalence of physical dysfunction in the population aged 45 and above is as high as 38.1%, of which 23.8% rely on others to assist with basic daily activities, and the prevalence of physical dysfunction in the population aged 80 and above has significantly increased. Physical dysfunction not only leads to an increase in the disability dependency rate and a decrease in quality of life but also directly increases medical expenses (with an average annual cost increase of 47%) and has been proven to be a strong predictor of all-cause mortality independent of chronic diseases (hazard ratio (HR)=2.18, 95% CI: 1.89–2.51), with a disease burden far exceeding traditional knowledge [3]. The

economic and social impacts of this phenomenon are profound and include labour shortages, a surge in pressure on the pension system, and the need for the health service system to adapt to high-demand populations that coexist with multiple illnesses.

In 2023, the American Heart Association (AHA) innovatively proposed the conceptual framework for cardiovascular kidney metabolic syndrome (CKM), which aims to integrate four disease entities that share a pathological basis: obesity, diabetes, chronic kidney disease, and cardiovascular disease. CKM syndrome is characterized by a vicious cycle of metabolic disorder, vascular damage, and declining organ function: insulin resistance promotes atherosclerosis, chronic inflammation accelerates the decline in renal function, and the decline in the estimated glomerular filtration rate (eGFR) further worsens metabolic homeostasis. The corresponding pathological and physiological core originates from dysfunction of adipose tissue, leading to inflammation, oxidative stress, and endothelial damage, and ultimately resulting in multiple organ failure [4]. The CKM burden of the Chinese population is particularly severe. Based on the joint analysis of CHARLS, Pinggu Research, and 3B Research, the prevalence rate of late CKM syndrome in men over 65 years old (stage 3–4) is 78.8%, which is significantly greater than that in women of the same age (44.6%, $P < 0.001$), highlighting the key role of sex differences [5]. This difference is partly attributed to higher rates of visceral obesity and decreased risk of low eGFR in males [6,7]. Notably, there is a bidirectional pathological association between CKM syndrome and physical dysfunction: arteriosclerosis (evaluated by pulse wave velocity [EPWV]) leads to insufficient tissue perfusion and induces muscle atrophy, and a decrease in eGFR promotes the accumulation of inflammatory factors such as IL-6 and TNF-α, activates the ubiquitin proteasome system, and accelerates muscle protein breakdown [4,8]. A team from Guangdong Provincial People's Hospital further reported that among the components of CKM, chronic kidney disease has the highest attributable risk for physical dysfunction (PAR% = 34.1%), whereas ischaemic heart disease leads to the most severe loss of healthy lifespan (YLDs = 643/100000) [8], indicating that CKM is not only a cluster of metabolic diseases but also a core driving factor for physical dysfunction.

Although the association between CKM and physical dysfunction has been partially elucidated, there are three significant limitations in current research. First, there is a lack of social dimensions: traditional biomedical models often overlook the core role of social determinants. Social isolation, a key indicator of social health, activates the hypothalamic pituitary adrenal axis (HPA axis), increases cortisol levels, and promotes insulin resistance and muscle breakdown metabolism [9,10]. Large cohort studies have confirmed that social isolation can increase the risk of physical dysfunction by 40%–60% (OR=1.52, 95% CI: 1.33–1.74), with an effect intensity comparable to that of hyperglycaemia (OR = 1.48) [11,12]. The unique changes in family structure in China (with empty nest families accounting for 25%) further exacerbate this risk, but the interaction effect between social isolation and CKD biomarkers, such as C-reactive protein (CRP) and interleukin-6 (IL-6), has not been systematically evaluated [10,13]. Second, the dynamic mechanism is unclear: CKM syndrome is a progressive process, but most studies rely on single time point measurements and cannot capture the cumulative effects of biomarkers. For example, long-term fluctuations in the atherogenic index (AIP) can predict cardiovascular events (AUC = 0.79), but similar dynamic models are still lacking in the field of dysfunction [14]. Finally, the mediating pathway has not been quantified: both arteriosclerosis (EPWV) and frailty (Frailty Index) have been shown to be independent predictors of physical dysfunction, but the causal pathway between the two and the moderating effect of social isolation remain unclear [15,16]. A study by Capital Medical University suggested that inflammation serves a mediating role between CKM and cognitive decline (accounting for 22%), but there is a lack of evidence regarding whether this mechanism extends to the field of physical dysfunction [17]. These limitations lead to insufficient identification and precise intervention strategies for high-risk populations, such as insufficient consideration of age/sex-specific effects (e.g., males under 60 years old are more susceptible to social isolation) [12,13].

To overcome the above limitations, this study innovatively constructed a "social biological" integrated analysis framework based on the nationally representative CHARLS 2015 database (N = 3756), aiming to address core scientific questions: Is social isolation independent of CKM markers (VAI, eGFR, and CKM staging) in predicting physical dysfunction?

Does arteriosclerosis (EPWV) affect physical dysfunction through the mediation of frailty? And is the cumulative exposure of CKM biomarkers (such as AIP) more predictive than a single measurement? [5,16]. Theoretical innovation is reflected in three aspects: conceptually, for the first time, social isolation (social dimension) and CKM syndrome (biological dimension) are integrated into a unified framework, verifying the hypothesis of "social psychological stress biomarker interaction" and filling the gap in research on the interaction between social factors and multiple system diseases [13]. Methodologically, causal mediation analysis (bootstrap = 1000) was introduced to analyse the pathway effects of "EPWV → frailty → physical dysfunction", and dynamic risk models were used to quantify the cumulative effects of biomarkers over time [18,19]. In terms of application, identifying high-risk subgroups (such as males under 60 years old and CKM stage 2 patients) provides targets for stratified interventions: for example, strengthening community support networks for socially isolated populations to block HPA axis activation and implementing early anti-frailty training for individuals with elevated EPWV to delay muscle breakdown [15,20]. This framework not only promotes the paradigm shift from "single-disease management" to "multisystem integrated intervention" but also provides an empirical basis for the precise prevention and control of physical dysfunction in a global ageing society.

## Materials and methods

### Study population

This study is based on publicly available data from the China Health and Retirement Longitudinal Study (CHARLS) database. The final sample size included in the analysis was 3756 participants, including 965 in the group without physical dysfunction and 2791 in the group with physical dysfunction. The study detailed the baseline demographic characteristics of the participants (such as age, sex, place of residence, marital status, and education level) and lifestyle information (such as smoking history and frequency of alcohol consumption) [21]. All analyses were conducted on the basis of the available variables in the database. This study used publicly available data from the 2015 nationwide follow-up wave of the China Health and Retirement Longitudinal Study (CHARLS). The data were obtained through authorization from the CHARLS data centre of Peking University on 19 May 2025. The original CHARLS study was approved by the Biomedical Ethics Review Committee of Peking University (IRB00001052−1015). The dataset used in this secondary analysis has been strictly de-identified and does not contain any personal identity information (such as name, ID number, or address). Researchers were unable to identify individual participants' identities throughout the entire process of data cleaning, analysis, and result reporting [22,23] (Fig 1).

### Variable definition and measurement

**Outcome variable. Physical dysfunction:** Calculation formula and assignment rules: The evaluation items include 9 basic physical function activities: running or jogging for 1 kilometre; walking 1 kilometre; walking 100 metres; rising after sitting for a long time; continuously climbing several layers of stairs; bending or squatting; raising arms over shoulders; picking up small coins from a table; and carrying heavy objects (>10 pounds). Answer options and coding: 0 for no difficulty, 1 for difficulty but still able to complete, 2 for difficulty and help required, and 3 for inability to complete. If a subject reported difficulty (coding ≥ 1) in any of the 9 items, this was defined as physical dysfunction [24].

**Core independent variables.** Social Isolation (SI): The construction in CHARLS research is based on five core items: marital status (1 point for unmarried, divorced, or widowed), living arrangement (1 point for living alone), social contact frequency with children (1 point for less than once a week), type of residence (1 point for rural areas), and participation in social activities (1 point for not participating in any social activities during the past month).

If each item met the conditions, it was given a score of 1 point. The total score ranges from 0 to 5 points. The higher the score is, the more serious the social isolation.

Finally, a total score of ≥ 2 points is considered as the critical point for judging social isolation (i.e., "yes"), and a total score of < 2 points is judged as non-social isolation ("no").

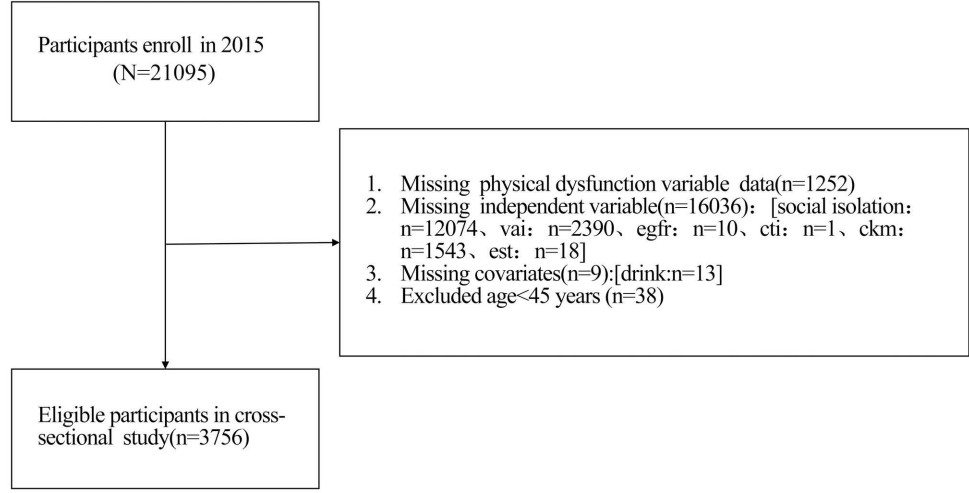

**Fig 1. Flow Diagram for Participant Inclusion in the Study.**

This scoring standard refers to the international health and retirement research (HRS) model, and each item is simply summed with equal weight (i.e., 1 point for each item) [25].

This critical value (≥ 2 points) has been verified in many international studies to effectively predict health outcomes such as mortality, cardiovascular disease, and chronic kidney disease. For example, a meta-analysis [26,27] revealed that social isolation increased the risk of all-cause death by 29% (OR = 1.29), coronary heart disease by 29%, and stroke by 32%.

In the Chinese population, the CHARLS cohort study (the sample covers middle-aged and elderly people in 28 provinces across the country) further confirmed [25] that the risk of renal function decline in social isolation with the same critical value (≥ 2 points) significantly increased, indicating that the standard has predictive validity for the Chinese population.

With respect to the processing of missing data, CHARLS routinely uses statistical techniques to fill in missing values (such as mode for classified variables and mean for continuous variables) and corrects sampling deviation by analysing weights to ensure national representativeness [28,29].

However, in the specific operation of the social isolation component, if the data of five core items are missing, this study chooses to directly eliminate the sample rather than to complete it.

This approach may reduce the sample size but avoid the classification bias that may be introduced by imputation, such as the lack of marital status or social activity participation. If the mode is filled, some actual isolators may be erroneously classified into a nonisolated group [25].

Of note [30], mentioned that other studies used different threshold values (such as ≥ 4 points to determine isolation), but CHARLS mainly used the international standard of ≥ 2 points and noted that the reliability and validity of this standard in the elderly population has been supported by many studies [31].

## Cardiovascular–kidney–metabolic (CKM) syndrome stages

Stage 0: no CKM risk factors (no overweight/obesity, metabolic abnormalities, CKD, or cardiovascular disease).

Stage 1: overweight/obesity (BMI ≥ 25 kg/m$^2$) or abdominal obesity but no other metabolic risk factors (such as hypertension or diabetes) or CKD.

Stage 2: presence of metabolic risk factors (hypertension, hypertriglyceridaemia, metabolic syndrome, diabetes) or CKD (eGFR < 60 mL/min/1.73 m$^2$ or proteinuria) but no cardiovascular disease.

Stage 3: at least one case of subclinical cardiovascular disease (such as atherosclerosis or subclinical heart failure) accompanied by overweight/obesity, metabolic risk factors, or CKD.

Stage 4: clinically diagnosed cardiovascular diseases (such as coronary heart disease, heart failure, and stroke) with at least one other risk factor (CKD, BMI ≥ 25 kg/m$^2$, diabetes, hypertension, etc.) are divided into no renal failure (4a) and renal failure (4b, eGFR < 15 mL/min/1.73 m$^2$ or renal replacement therapy) [32,33].

## Vascular ageing biomarkers

***Estimated pulse wave velocity (EPWV).*** EPWV is calculated on the basis of the MBP and age using the following formula: EPWV $= 9.587 - (0.402 \times age) + (4.560 \times 10^{-3} \times age^2) - (2.621 \times 10^{-5} \times age^2 \times MBP) + (3.176 \times 10^{-3} \times age \times MBP) - (1.832 \times 10^{-2} \times MBP)$, where MBP $= DBP + 0.4 (SBP - DBP)$ [34].

Body Composition Indicators:

Accessory skeletal muscle mass (ASM):

ASM $= 0.193 \times weight(kg) + 0.107 \times height(cm) - 4.157 \times gender - 0.037 \times age - 2.631$ Male sex = 1, female sex = 2, sum of skeletal muscle mass in arms and legs (kg), and assessment of muscle mass [35].

Frailty Index $= \frac{\text{Number of health deficits present}}{\text{Total number of deficits assessed}}$, which is a comprehensive assessment of individual physiological reserve decline and vulnerability [36].

Frailty: Using a Fried phenotypic model, weakness was defined as a score ≥ 3 for the following five criteria: involuntary weight loss (weight loss ≥ 4.5 kg or 5% in the past year), self-reported fatigue (CES-D assessment), a decrease in grip strength (BMI and sex correction), slow walking speed (4-metre walking test), and low physical activity (IPAQ questionnaire assessment) [37].

When scoring, each criterion is a binary variable (existence = 1 point; nonexistence = 0 points). If the total score is ≥ 3 points, it is judged as frailty, and if the total score is < 3 points, it is judged as non-frailty. (yes/no) [38].

Visceral Adiposity Index (VAI):

$$VAI_{men} = \left( \frac{\text{Waist circumference}}{39.68 + (1.88 \times BMI)} \right) \times \left( \frac{\text{Triglycerides}}{1.03} \right) \times \left( \frac{1.31}{HDL - C} \right)$$

$$VAI_{women} = \left( \frac{\text{Waist circumference}}{36.58 + (1.89 \times BMI)} \right) \times \left( \frac{\text{Triglycerides}}{0.81} \right) \times \left( \frac{1.52}{HDL - C} \right)$$

Indicators reflecting visceral fat function and quantity were calculated on the basis of waist circumference, body mass index, triglyceride levels, and high-density lipoprotein cholesterol levels [39].

## Core indicators of renal function

Estimated Glomerular Filtration Rate, eGFR:
Female:

- Scr $\leq 62$ μmol/L $\rightarrow$ eGFR $= 144 \times (Scr/62)^{-0.329} \times 0.993 Age$

- Scr $> 62$ μmol/L $\rightarrow$ eGFR $= 144 \times (Scr/62)^{-1.209} \times 0.993 Age$

Male:

- Scr $\leq 80$ μmol/L $\rightarrow$ eGFR $= 141 \times (Scr/80)^{-0.411} \times 0.993 Age$

- Scr $> 80$ μmol/L $\rightarrow$ eGFR $= 141 \times (Scr/80)^{-1.209} \times 0.993 Age$

followed by the assessment of renal function [40].

### Lipid metabolism indicators

C-reactive protein triglyceride glucose index (CTI): The formula for detecting C-reactive protein, triglyceride, and glucose levels in blood samples is as follows:

$$CTI = 0.412 \times \ln(C-\text{reactive protein [CRP]}) + \ln(\text{triglycerides [mg/dL]} \times \text{fasting blood glucose [mg/dL]}/2)$$ [41]. As a novel composite biomarker that integrates inflammation and lipid metabolism, CTI has shown significant potential in predicting neurodegenerative diseases [42].

**Covariates.** Demographic factors: age (binary classification: ≥ 60 years/ < 60 years) [43] and sex [44].

Behavioural factors: smoking history (NO/YES) [45] and drinking frequency (low frequency/intermediate frequency/high frequency) [46].

Social background: place of residence (urban/rural) [47], marital status (unmarried/married) [48], and education level (no formal education/high school or belt/above high school) [49].

### Statistical analysis

Descriptive statistics were used to summarize the demographic, behavioural, and clinical characteristics and biomarker profiles of the study participants. Continuous variables with a normal distribution are presented as the mean ± standard deviation (SD), whereas categorical variables are expressed as frequencies and percentages (%). To compare baseline characteristics between groups with and without physical dysfunction, Pearson's chi-square tests were employed for categorical variables, and independent Student's t tests or one-way analysis of variance (ANOVA) were used for continuous variables, as appropriate [50,51].

All the statistical analyses were performed using R software (version 4.2.2). A two-sided p value < 0.05 was considered to indicate statistical significance for all tests.

The associations between core exposures (social isolation and multidimensional biomarkers, including VAI, eGFR, frailty index, ASM, CKM syndrome stage, and EPWV) and physical dysfunction (binary outcome) were assessed using multivariable logistic regression. The results are reported as odds ratios (ORs) with 95% confidence intervals (CIs). Two adjusted models were constructed: Model I was adjusted for basic demographic factors (age and sex), and Model II was further adjusted for lifestyle factors (smoking status and alcohol consumption) and socioeconomic factors (education level, marital status, and residential area) [52,53].

Stratification analyses were conducted to examine potential effect modifications for key subgroups, including age (<60 vs. ≥ 60 years), sex, residential area, marital status, education level, smoking history, and drinking frequency. Interaction terms were introduced into the logistic models, and subgroup-specific ORs were reported [54].

To explore the potential nonlinear relationships between continuous biomarkers (VAI, eGFR, frailty index, ASM, CTI, and EPWV) and physical dysfunction, a preliminary analysis was conducted by smooth curve fitting after adjusting for covariates [55]. If the visual inspection indicates that there is a nonlinear correlation, the two piecewise linear regression models [56] are applied to identify the inflection points through the iterative method system. The determination of the optimal threshold is based on the principle of maximizing the goodness of fit of the model, and the log likelihood ratio test is used to verify the statistical significance of the threshold effect. This method does not rely on a treatment group/control group design and is suitable for observational data analysis [57]. Its goal is to realize the objective determination and verification of critical biomarker values through the combination of nonlinear modelling and statistical tests.

Causal mediation analysis was carried out using the R package "mediation" with 1000 bootstrap simulations to estimate the direct, indirect, and total effects of key pathways—for instance, the mediating role of the frailty index in the association between EPWV and physical dysfunction. All the mediation models were adjusted for the full set of covariates (age, sex, residence, marital status, education, smoking status, and drinking status). The mediation proportion was calculated as the

ratio of the indirect effect to the total effect. Bias-corrected bootstrap confidence intervals were reported for the mediation effects [50,51].

### Sensitivity analysis for threshold effects

To assess the robustness of the identified threshold associations to potential unmeasured confounding, we calculated E-values for the significant segments of the piecewise linear models [58]. The E-value quantifies the minimum strength of association that an unmeasured confounder would need to have with both the exposure and the outcome, conditional on the measured covariates, to fully explain away the observed exposure-outcome association [59]. We report both the point estimate of the E-value for the observed effect size and the E-value for the lower limit of the 95% confidence interval. Larger E-values indicate that stronger unmeasured confounding would be required to nullify the association, thus implying greater robustness of the finding. The detailed results of the E-value sensitivity analysis for all biomarkers are presented in Supplementary Table 1.

## Results

### Baseline characteristics

Ultimately, 3756 participants aged ≥ 45 years were included (Table 1). According to the physical dysfunction status stratification, there were 2791 individuals in the physical dysfunction group (PD) and 965 individuals in the nonphysical dysfunction group (without PD). There were significant differences in multiple baseline characteristics between the two groups: the sociodemographic characteristics were as follows: the proportion of participants aged 60 years or older in the physical dysfunction group was significantly greater (66.14% vs. 50.78%, $P<0.001$), the proportion of females was greater (54.14% vs. 31.30%, $P<0.001$), the proportion of rural residents was greater (71.37% vs. 66.43%, $P=0.004$), the proportion of unmarried individuals was greater (24.87% vs. 14.30%, $P<0.001$), and the proportion of participants who had not received formal education was greater (50.91% vs. 28.81%, $P<0.001$). In addition, the prevalence of social isolation was significantly greater in the physical dysfunction group (38.95% vs. 25.60%, $P<0.001$). Lifestyle: The proportion of current smokers in the physical dysfunction group was lower (44.64% vs. 58.24%, $P<0.001$), and the proportion of high-frequency drinkers was lower (12.76% vs. 22.38%, $P<0.001$). Health indicators and biomarkers: The physical dysfunction group showed a comprehensive deterioration of biomarker levels, with a significantly higher frailty index (6.905 ± 4.028 vs. 2.348 ± 1.437, $P<0.001$), significantly higher EPWVs (10.239 ± 1.971 vs. 9.492 ± 1.772 m/s, $P<0.001$), significantly lower eGFR (87.427 ± 16.179 vs. 90.296 ± 15.853 mL/min/1.73 m², $P<0.001$), significantly higher VAI (4.919 ± 4.300 vs. 4.386 ± 4.326, $P<0.001$), significantly lower ASM (16.767 ± 4.163 vs. 18.707 ± 3.884, $P<0.001$), and significantly higher CTI (8.812 ± 0.854 vs. 8.715 ± 0.854, $P<0.001$). Clinical status: The proportion of CKM (stage 4) patients in the physical dysfunction group was significantly greater (27.88% vs. 10.05%, $P<0.001$), and the proportion of frail patients in this group was 31.42%, while the nonphysical dysfunction group had no cases of frailty.

### Multivariate regression analysis

According to the multivariate regression model (Adjusted II) with all covariates adjusted (Table 2), social isolation significantly increased the risk of physical dysfunction (aOR=1.380; 95% CI: 1.132–1.683; $P=0.002$). CKM syndrome stage was significantly correlated with risk, with risk increasing with stage (stage 3 vs. stage 0: aOR=1.363; 95% CI: 0.785–2.364; $P=0.271$; stage 4 vs. stage 0: aOR=4.805; 95% CI: 2.691–8.579; $P<0.001$). Vascular and metabolic markers: EPWV: aOR=1.252, 95% CI: 1.195–1.312, $P<0.001$; frailty index: aOR=2.190, 95% CI: 2.060–2.329, $P<0.001$, eGFR: aOR=0.990, 95% CI: 0.985–0.995, $P<0.001$; ASM: aOR=0.936, 95% CI: 0.915–0.958, $P<0.001$; CTI: aOR=1.134, 95% CI: 1.036–1.241, $P=0.007$; and VAI: aOR=1.015, 95% CI: 0.996–1.035, $P=0.120$.

### Threshold effect analysis

The threshold effect model revealed a nonlinear relationship between multiple exposure factors and the risk of physical dysfunction (Table 3 and Fig 2). The key findings are as follows: EPWV: There is a significant inflection point (K=7.178

**Table 1. Baseline Characteristics of the Population Stratified by PD (n = 3756).**

| Characteristics | Without PD | With PD | P value |
|---|---|---|---|
| N | 965 | 2791 | |
| **AGE** | | | <0.001 |
| <60 | 475 (49.223%) | 945 (33.859%) | |
| ≥60 | 490 (50.777%) | 1846 (66.141%) | |
| **Gender** | | | <0.001 |
| Male | 663 (68.705%) | 1280 (45.862%) | |
| Female | 302 (31.295%) | 1511 (54.138%) | |
| **Living Area** | | | 0.004 |
| Urban Community | 324 (33.575%) | 799 (28.628%) | |
| Rural Village | 641 (66.425%) | 1992 (71.372%) | |
| **Married Status** | | | <0.001 |
| Unmarried | 138 (14.301%) | 694 (24.866%) | |
| Married | 827 (85.699%) | 2097 (75.134%) | |
| **Education** | | | <0.001 |
| No formal education | 278 (28.808%) | 1421 (50.914%) | |
| High school and below | 625 (64.767%) | 1297 (46.471%) | |
| Above high school | 62 (6.425%) | 73 (2.616%) | |
| **Smoking status** | | | <0.001 |
| NO | 403 (41.762%) | 1545 (55.357%) | |
| YES | 562 (58.238%) | 1246 (44.643%) | |
| **Drinking Status** | | | <0.001 |
| Low frequency | 679 (70.363%) | 2291 (82.085%) | |
| Intermediate frequency | 70 (7.254%) | 144 (5.159%) | |
| High frequency | 216 (22.383%) | 356 (12.755%) | |
| **Social Isolation** | | | <0.001 |
| NO | 718 (74.404%) | 1704 (61.053%) | |
| YES | 247 (25.596%) | 1087 (38.947%) | |
| **Frailty** | | | <0.001 |
| NO | 965 (100.000%) | 1914 (68.578%) | |
| YES | 0 (0.000%) | 877 (31.422%) | |
| **CKM** | | | <0.001 |
| Phase 0 | 21 (2.176%) | 47 (1.684%) | |
| Phase 1 | 70 (7.254%) | 170 (6.091%) | |
| Phase 2 | 167 (17.306%) | 470 (16.840%) | |
| Phase 3 | 610 (63.212%) | 1326 (47.510%) | |
| Phase 4 | 97 (10.052%) | 778 (27.875%) | |
| **VAI** | 4.386 ± 4.326 | 4.919 ± 4.300 | <0.001 |
| **eGFR** | 90.296 ± 15.853 | 87.427 ± 16.179 | <0.001 |
| Frailty Index | 2.348 ± 1.437 | 6.905 ± 4.028 | <0.001 |
| **ASM** | 18.707 ± 3.884 | 16.767 ± 4.163 | <0.001 |
| **CTI** | 8.715 ± 0.864 | 8.812 ± 0.854 | 0.003 |
| Estimated pulse wave velocity | 9.492 ± 1.772 | 10.239 ± 1.971 | <0.001 |

The data is expressed as mean ± standard deviation or frequency (percentage).

**Table 2. Multiple Logistic Regression Equation.**

| Exposure | Nonadjusted | Adjusted I | Adjusted II |
|---|---|---|---|
| **Social Isolation** | | | |
| NO | 1 | 1 | 1 |
| YES | 1.854 (1.575, 2.183) <0.001 | 1.474 (1.241, 1.752) 0.001 | 1.380 (1.132, 1.683) 0.002 |
| **VAI** | 1.032 (1.013, 1.052) 0.001 | 1.007 (0.988, 1.027) 0.455 | 1.015 (0.996, 1.035) 0.120 |
| **eGFR** | 0.988 (0.983, 0.993) <0.001 | 0.995 (0.990, 1.000) 0.073 | 0.990 (0.985, 0.995) <0.001 |
| **Frailty Index** | 2.216 (2.087, 2.354) <0.001 | 2.174 (2.045, 2.311) <0.001 | 2.190 (2.060, 2.329) <0.001 |
| **Frailty** | | | |
| NO | 1 | 1 | 1 |
| YES | NA | NA | NA |
| **ASM** | 0.892 (0.876, 0.909) <0.001 | 0.974 (0.946, 1.002) 0.069 | 0.936 (0.915, 0.958) <0.001 |
| **CTI** | 1.143 (1.048, 1.247) 0.003 | 1.102 (1.007, 1.205) 0.034 | 1.134 (1.036, 1.241) 0.007 |
| **CKM** | | | |
| Phase 0 | 1 | 1 | 1 |
| Phase 1 | 1.085 (0.605, 1.948) 0.784 | 1.113 (0.611, 2.027) 0.727 | 0.991 (0.543, 1.809) 0.977 |
| Phase 2 | 1.257 (0.730, 2.166) 0.409 | 1.305 (0.747, 2.280) 0.349 | 1.235 (0.706, 2.161) 0.459 |
| Phase 3 | 0.971 (0.576, 1.639) 0.913 | 1.968 (1.117, 3.467) 0.019 | 1.363 (0.785, 2.364) 0.271 |
| Phase 4 | 3.584 (2.055, 6.249) <0.001 | 5.531 (3.066, 9.977) <0.001 | 4.805 (2.691, 8.579) <0.001 |
| Estimated pulse wave velocity | 1.236 (1.186, 1.287) <0.001 | 1.233 (1.165, 1.304) <0.001 | 1.252 (1.195, 1.312) <0.001 |

Data in the table: β (95%CI) Pvalue/ OR (95%CI) Pvalue.

Result variable:Physical dysfunction.

Exposed variables:Social Isolation;VAI;eGFR;Frailty Index;Frailty;ASM;CTI;CKM;Estimated pulse wave velocity.

Non-adjusted model adjust for: None.

Adjust I model adjust for: AGE; Gender.

Adjust II model adjust for:Living Area;Married Status;Education;Smoking status; Drinking Status.

**Table 3. Threshold Effect Analysis.**

| Exposure: | VAI | eGFR | Frailty Index | ASM | CTI | ESTIMATED PULSE WAVE VELOCITY |
|---|---|---|---|---|---|---|
| Outcome: Physical Dysfunction | OR (95% CI) P value | OR (95% CI) P value | OR (95% CI) P value | OR (95% CI) P value | OR (95% CI) P value | OR (95% CI) P value |
| Straight-line effect | 1.008 (0.989, 1.029) 0.405 | 0.995 (0.990, 1.001) 0.077 | 2.167 (2.037, 2.305) <0.001 | 1.001 (0.971, 1.031) 0.967 | 1.122 (1.024, 1.230) 0.013 | 1.243 (1.172, 1.319) <0.001 |
| Model II | 1.008 (0.989, 1.029) 0.405 | 0.995 (0.990, 1.001) 0.077 | 2.167 (2.037, 2.305) <0.001 | 1.001 (0.971, 1.031) 0.967 | 1.122 (1.024, 1.230) 0.013 | 1.243 (1.172, 1.319) <0.001 |
| Folding point (K) | 13.053 | 70.958 | 7.679 | 22.94 | 10.348 | 7.178 |
| <K-segment effect 1 | 1.032 (1.002, 1.063) 0.038 | 1.006 (0.993, 1.019) 0.384 | 2.124 (1.993, 2.264) <0.001 | 0.972 (0.938, 1.008) 0.124 | 1.182 (1.066, 1.311) 0.002 | 2.026 (1.113, 3.690) 0.021 |
| >K-segment effect 2 | 0.964 (0.920, 1.009) 0.112 | 0.990 (0.982, 0.998) 0.013 | NA | 1.166 (1.041, 1.307) 0.008 | 0.586 (0.319, 1.078) 0.086 | 1.226 (1.153, 1.304) <0.001 |
| Log Likelihood Ratio Tests | 0.041 | 0.082 | 0.009 | 0.003 | 0.040 | 0.110 |

Data in the table: β (95%CI) Pvalue/ OR (95%CI) Pvalue.

Result variable:Physical dysfunction.

Exposed variables:VAI;eGFR;Frailty Index;ASM;CTI;Estimated pulse wave velocity.

Adjust variables:AGE;Gender;Living Area;Married Status;Education;Smoking status;Drinking Status.

**Fig 2. Smooth Curve Fitting.**

m/s). When the EPWV was<7.178 m/s, for every 1 unit increase, the risk of physical dysfunction significantly increased by 102.6% (OR=2.026; 95% CI: 1.113–3.690; *P*=0.021). When the EPWV was ≥ 7.178 m/s, for every 1-unit increase, the risk of physical dysfunction significantly increased by 22.6% (OR=1.226; 95% CI: 1.153–1.304; *P*<0.001). FRAILTY INDEX: There is also a significant inflection point (K=7.679), and the log likelihood ratio test result is significant (*P*=0.009). When

the frailty index is less than 7.679, for every 1 unit increase, the risk of physical dysfunction significantly increases by 112.4% (OR=2.124; 95% CI: 1.993–2.264; $P<0.001$). Other exposure factors: VAI: inflection point K=13.053. The risk increased before the inflection point (OR=1.032; $P=0.038$) and decreased after the inflection point (OR=0.964; $P=0.112$), and the overall threshold effect test was significant ($P=0.041$). eGFR: inflection point K=70.958. The preinflection point effect was not significant (OR=1.006; $P=0.384$), whereas the postinflection point risk was significantly reduced (OR=0.990; $P=0.013$). The overall threshold effect test revealed a significant marginal effect ($P=0.082$). ASM: inflection point K=22.94. The preinflection point effect was not significant (OR=0.972; $P=0.124$), whereas the postinflection point risk significantly increased (OR=1.166; $P=0.008$). The overall threshold effect test was significant ($P=0.003$). CTI: inflection point K=10.348. The risk significantly increased before the inflection point (OR=1.182; $P=0.002$), but the trend of risk reduction after the inflection point was not significant (OR=0.586; $P=0.086$). The overall threshold effect test was significant ($P=0.040$).

Smooth curve fitting relationships were determined between (A) the VAI and physical dysfunction, (B) the eGFR and physical dysfunction, (C) the frailty index and physical dysfunction, (D) the ASM and physical dysfunction, (E) the CTI and physical dysfunction, and (F) the estimated pulse wave velocity and physical dysfunction.

## Subgroup analysis

Subgroup analysis revealed effect modification (Fig 3), indicating that the effect of social isolation on physical dysfunction varies across different subgroups of the population. Stratified by age, social isolation significantly increased the risk of physical dysfunction in individuals under the age of 60 (OR=1.731, 95% CI: 1.229–2.436; $P=0.002$). There was no statistically significant association between social isolation and physical dysfunction in individuals aged 60 years or older (OR=1.146, 95% CI: 0.889–1.477; $P=0.294$). Stratified by sex, male social isolation significantly increased the risk of physical dysfunction (OR=1.400, 95% CI: 1.089–1.800; $P=0.009$). There was no statistically significant association between female social isolation and physical dysfunction (OR=1.219, 95% CI: 0.864–1.722; $P=0.260$). Stratified by the place of residence, social isolation significantly increased the risk of physical dysfunction among rural residents (OR=1.679, 95% CI: 1.138–2.476; $P=0.009$). There was no statistically significant association between social isolation and physical dysfunction among urban residents (OR=1.215, 95% CI: 0.957–1.542; $P=0.109$). Stratified by marital status, a marginally significant association was observed between social isolation and an increased risk of physical dysfunction only in the married group (OR=1.289, 95% CI: 1.035–1.605; $P=0.024$), and not in the single group (OR=1.541, 95% CI: 0.924–2.571; $P=0.098$). Stratified by educational level, social isolation significantly increased only the risk of physical dysfunction in the high school and below groups (OR=1.362, 95% CI: 1.044–1.776; $P=0.023$). There was no significant association between the no formal education group and the above high school group (OR=1.238; 95% CI: 0.895–1.712; $P=0.198$; OR=3.813; 95% CI: 0.832–17.476; $P=0.085$). Stratified by smoking history, social isolation significantly increased only the risk of physical dysfunction in the former smoking group (OR=1.328, 95% CI: 1.012–1.742; $P=0.041$). The association margin was significant in the nonsmoking group (OR=1.340, 95% CI: 0.987–1.821; $P=0.061$). After stratification by alcohol consumption frequency, social isolation significantly increased the risk of physical dysfunction only in the low-frequency group (OR=1.381, 95% CI: 1.090–1.751; $P=0.008$). There was no significant association between the intermediate-frequency group and the high-frequency group (OR=0.999, 95% CI: 0.431–2.319, $P=0.999$; OR=1.288, 95% CI: 0.819–2.024, $P=0.273$). Notably, the exposure variable EPWV (estimated pulse wave velocity) was significantly positively correlated with the risk of physical dysfunction in all the subgroups of stratified variables (age, sex, place of residence, marital status, education level, smoking history, and frequency of alcohol consumption) (all $P<0.001$), indicating that there is no significant heterogeneity in the impact of EPWV on physical dysfunction among the different subgroups.

 

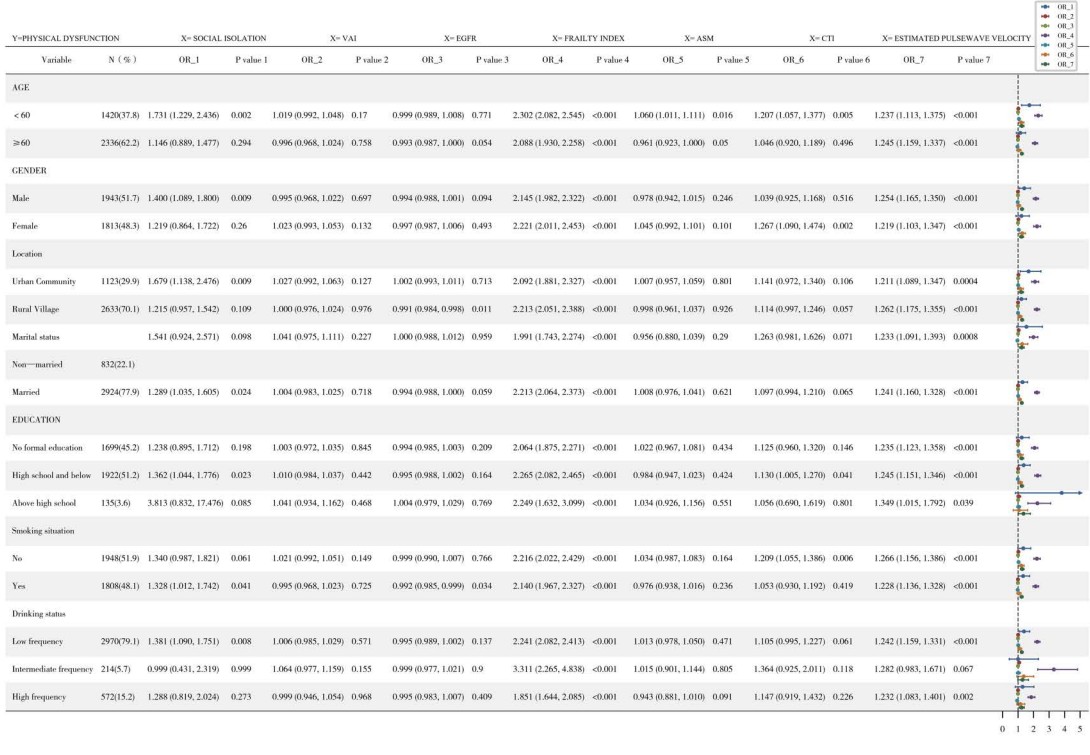

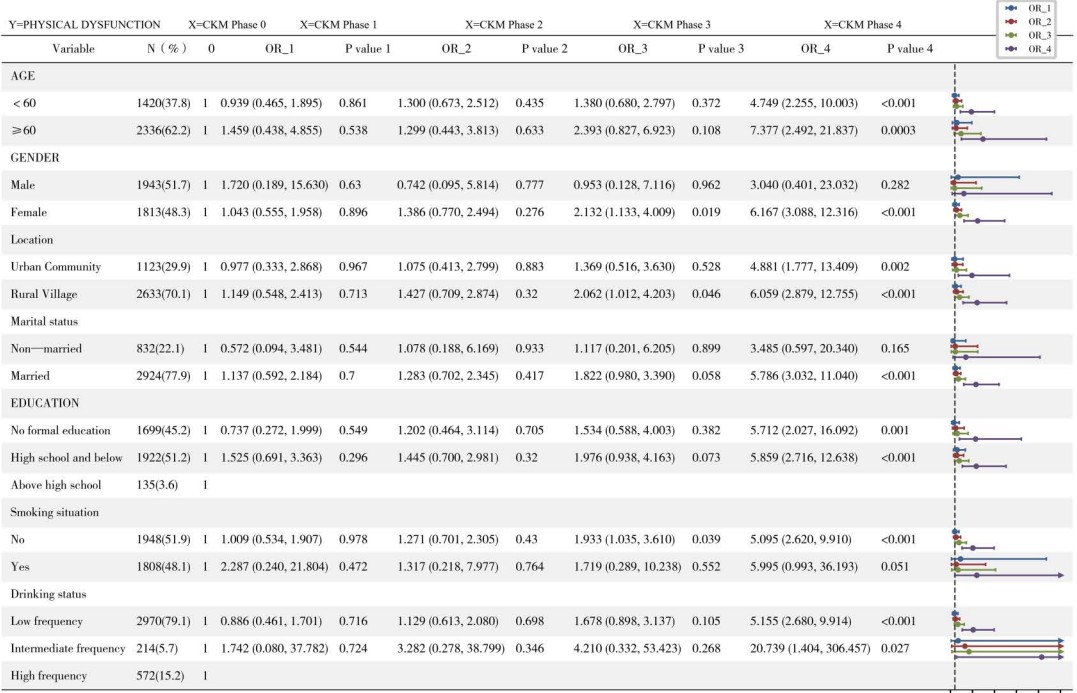

**Fig 3. Subgroup Analysis.**

## Causal mediation analysis

In this study, causal mediation analysis was conducted using the nonparametric bootstrap method (1000 resamplings) (Fig 4), with a sample size of n = 3756. Adjusted variables included age, sex, place of residence, marital status, education level, smoking history, and frequency of alcohol consumption. The key findings are as follows: EPWV → frailty index → physical dysfunction pathway: total effect: β = 0.090 (95% CI: 0.067–0.116; $P<0.001$), mediation effect: β = 0.052 (95% CI: 0.040–0.064; $P<0.001$), and mediation effect ratio: 57.8% (95% CI: 44.3–76.5%; $P<0.001$).

EPWV → Social Isolation → Physical Dysfunction pathway: total effect: β = 0.089 (95% CI: 0.066–0.115; $P<0.001$), mediating effect: β = 0.001 (95% CI: 0.0001–0.004; $P=0.028$), and mediating effect ratio: 0.9% (95% CI: 0.1–4.6%; $P=0.028$). The other pathway analysis results were as follows: the frailty index → EPWV → physical dysfunction: the mediation ratio was only 1.6% (95% CI: 0.7–2.6%; $P=0.002$); CKM (cardiovascular metabolic health) → EPWV → physical dysfunction: the mediation ratio was 13.4% (95% CI: 7.7–19.8%; $P<0.001$).

## Discussion

This study revealed that social isolation independently increases the risk of physical dysfunction by 38% (aOR=1.380) among ageing Chinese adults, with amplified effects in males, rural residents, and those aged <60 years. These findings align with global evidence indicating that social isolation is a robust predictor of functional decline. For instance, Mehrabi and Béland [60] reported that social isolation is a significant risk factor for frailty-related disability, particularly among community-dwelling older adults with preexisting health vulnerabilities [61]. Our results extend this understanding by

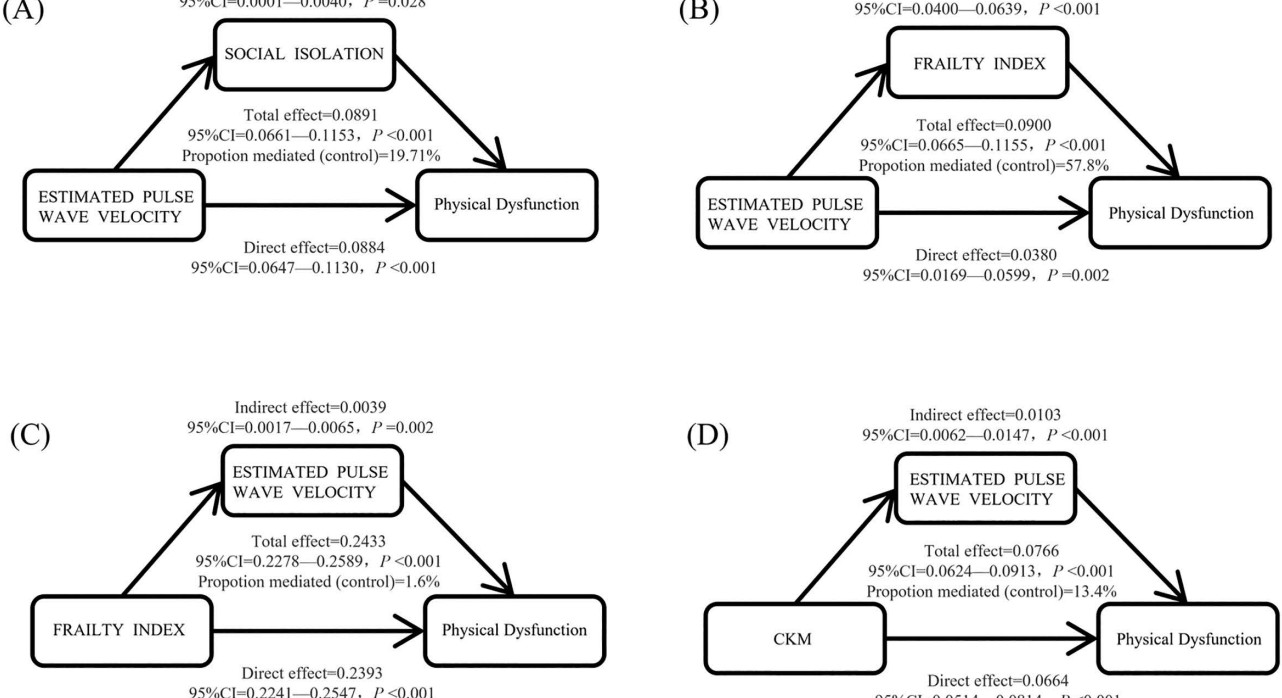

**Fig 4. Causal Mediation Analysis.** (A) Mediating effect analysis of social isolation as an indicator of estimated pulse wave velocity as it relates to physical dysfunction. (B) Mediating effect analysis of the frailty index as an indicator of estimated pulse wave velocity as it relates to physical dysfunction. (C) Mediating effect analysis of estimated pulse wave velocity as an indicator of the frailty index as it relates to physical dysfunction. (D) Mediating effect analysis of the estimated pulse wave velocity as an indicator of the CKM as it relates to physical dysfunction.

highlighting demographic disparities within the Chinese context, suggesting that sociocultural and environmental factors may modulate the impact of isolation.

The synergistic effects of advanced CKM syndrome (Stage 4) and social isolation on physical dysfunction—yielding a 4.8-fold risk increase—underscore the critical interplay between biological and social determinants. This aligns with Fried et al.'s [18] phenotype model, which posits that frailty arises from multisystem dysregulation (e.g., metabolic, cardiovascular) and is exacerbated by psychosocial stressors. Our identification of nonlinear thresholds, such as the EPWV inflection point at 7.178 m/s (102.6% risk increase per unit below the threshold), further refines this model. These thresholds offer clinically actionable targets, in contrast to earlier studies that treated vascular ageing as a linear continuum without critical risk boundaries [18,62].

Frailty mediated 57.8% of the effect of vascular stiffness (EPWV) on physical dysfunction, establishing it as a pivotal pathway. This mediation effect exceeds the rates reported in similar studies: for example, Bai et al. [63] reported that depressive symptoms mediated only 18.5% of the social isolation–cognitive frailty relationship. Our higher mediation percentage suggests that frailty may be a more central biological conduit than psychological factors in terms of physical (vs. cognitive) decline. Rockwood's frailty index framework supports this, emphasizing cumulative deficits across physiological systems as drivers of functional impairment [64].

The identified ASM threshold (22.94 kg) for dysfunction risk corroborates Fried's frailty criterion for muscle weakness but provides a quantitative benchmark for intervention [18]. Notably, our muscle mass threshold diverges from that in Western populations (e.g., Fried's cohort), likely reflecting ethnic differences in body composition—a gap in prior frailty phenotyping [18,62]. Additionally, the stronger isolation effects in younger participants (<60 years) challenge the assumption that frailty mechanisms are uniform across age groups, highlighting the need for life-course approaches [65].

While the substantial mediating effect of frailty is a key finding, its interpretation warrants caution. Although our frailty index (a cumulative deficit measure) and the outcome of physical dysfunction (difficulty in ADLs) are conceptually and operationally distinct—with ADL items not included in the construction of the frailty index—we cannot fully rule out the possibility of some conceptual overlap between general vulnerability and specific functional limitations [66]. This potential overlap might lead to an overestimation of the mediation effect. Nevertheless, the strong association underscores the pivotal role of overall physiological decline in the pathway from vascular ageing to functional disability [67,68].

Furthermore, our cross-sectional design limits causal inference regarding the directionality between vascular ageing and frailty [67]. The relationship is likely bidirectional. Elevated EPWV may contribute to frailty by impairing microcirculation and promoting sarcopenia and chronic inflammation. Conversely, the proinflammatory and catabolic state characteristics of frailty can accelerate arterial stiffening and endothelial dysfunction, creating a vicious cycle [69]. Future longitudinal studies employing cross-lagged panel analyses are essential to elucidating the temporal precedence and bidirectional nature of this relationship.

A further limitation stems from the use of data from 2015. China's rapid societal evolution means that changes in social structures (e.g., urbanization and migration patterns), health care access, and medical management of chronic conditions might affect the generalizability of our exact estimates [69]. For instance, improved health care might alter the progression from CKM syndrome to dysfunction, potentially modifying the effect sizes we observed. However, we posit that the fundamental biological pathways linking social isolation, vascular ageing, frailty, and physical function are likely to remain valid. The key thresholds identified (e.g., for EPWV) may still serve as valuable risk indicators, although their precise values should be confirmed in more contemporary cohorts [70,71].

Our subgroup analyses revealed a particularly strong association between social isolation and physical dysfunction among males under 60 years of age [72,73]. These intriguing findings warrant further exploration beyond biological explanations. We hypothesize that sociocultural and economic factors may underpin this effect modification. First, gender norms in traditional Chinese society often emphasize men as primary breadwinners and pillars of strength. Social isolation among middle-aged men, which may stem from unemployment or workplace difficulties, could represent a profound

failure to meet these societal expectations, leading to greater psychological distress and faster health deterioration in middle-aged men than in women or older men, who may have different social roles and support expectations [74]. Second, occupational patterns are central to the identity and social networks of middle-aged men. Disruption in this domain (e.g., job loss, retirement) can be a potent trigger of both social isolation and functional decline [73]. Finally, while not directly measured, the migration history prevalent in rural China may also play a role: male migrants often experience fractured social networks because of mobility, making them more vulnerable to the health impacts of isolation upon return or in unstable work environments [74]. These nonbiological pathways offer a plausible framework for understanding the heightened vulnerability observed in this demographic and highlight the need for targeted interventions that address the unique psychosocial challenges faced by middle-aged men [72,73].

A key strength of our analysis is the evaluation of the robustness of the identified biomarker thresholds to potential unmeasured confounding. We employed E-values to quantify the strength of association an unmeasured confounder would need to have with both the biomarker and physical dysfunction to explain away the observed relationships [58]. For the significant threshold associations—such as the effect of EPWV below 7.178 m/s (OR=2.026, E-value = 2.20) and the frailty index below 7.679 (OR=2.124, E-value = 2.27)—the E-values were substantively greater than 1. This indicates that to fully nullify these associations, an unmeasured confounder would need to be associated with both the exposure and the outcome by risk ratios of 2.20-fold and 2.27-fold, respectively, which is a relatively strong magnitude for common confounders. Similarly, the E-values for other biomarkers like ASM (E-value = 1.37) and CTI (E-value = 1.40) also suggest a degree of robustness, as they exceed the strengths of many measured confounders in epidemiological studies. While this does not preclude the possibility of residual confounding, it increases confidence that the non-linear relationships we identified are not easily attributable to a single, strong, unmeasured confounder.

Despite adjusting for a comprehensive set of demographic, lifestyle, and socioeconomic covariates, our study is susceptible to residual confounding, an inherent limitation of observational cross-sectional designs [75,76]. Unmeasured factors such as depressive symptoms, detailed patterns of physical activity, and nutritional status may act as potential confounders. Depressive symptoms can lead to social withdrawal while simultaneously influencing physical function through inflammatory pathways [77]. Similarly, poor nutrition and sedentary behaviour are risk factors for both social isolation and physical decline [78]. Although the CHARLS database contains some related variables (e.g., CES-D for depression), detailed measures of physical activity intensity and dietary quality were not fully available or adjusted for in this analysis [79]. It is possible that the observed associations between social isolation and physical dysfunction could be partly influenced by these factors [80]. Future longitudinal studies incorporating more precise measurements of psychological health, objective physical activity, and nutritional intake are warranted to corroborate our findings and better disentangle these complex relationships. Limitations and research gaps: while CHARLS data offer national representativeness, the cross-sectional design limits causal inference. Longitudinal studies, such as Fried's 7-year follow-up, are needed to validate the identified mediation pathways [18]. Our focus on physical dysfunction also warrants comparisons with cognitive outcomes, as Bai et al. [63] demonstrated isolation's distinct pathways with respect to cognitive frailty. Future trials should test interventions targeting our identified thresholds (e.g., EPWV <7.178 m/s) to disrupt the isolation–frailty–dysfunction cascade.

## Conclusions

This study suggests that the synergistic effects of social isolation and multidimensional biomarkers (vascular function, metabolic status, muscle mass, and degree of weakness) are the core driving mechanism for physical dysfunction in the middle-aged and elderly population in China. Social isolation induces chronic stress and the accumulation of inflammation and accelerates the process of biomarker decline and deterioration, ultimately forming a vicious cycle of "isolation–comorbidity–disability" together with cardiovascular renal metabolic syndrome. Among them, frailty, as a key intermediary hub, significantly amplifies the damaging effects of pathological changes such as vascular ageing on function, and the

discovery of specific biological thresholds (such as arterial stiffness inflection points and lower limits of muscle mass protection) provides an important window for early intervention. These findings emphasize that implementing a strategy that integrates social network reconstruction, biomarker screening, and comorbidity management is key to blocking the chain reaction of physical dysfunction. In the future, in-depth analysis of the dynamic interaction mechanisms of multidimensional factors through longitudinal research is needed to provide a basis for precise intervention.

## Supporting information

**S1 Data.  Raw data.** The raw data mentioned in this article are all located in this file. (XLSX)

**S1 File.  README.** The data interpretation of the raw data mentioned in this article is located in this file. (DOCX)

**S1 Table.  The table presents the results of the sensitivity analysis conducted to assess the potential impact of a threshold effect.** (DOCX)

## Acknowlegment

The authors wish to express their sincere gratitude to all those who provided intellectual inspiration, technical support, and moral encouragement throughout the course of this study.

## Author contributions

**Conceptualization:** Shuang Deng, Zhongqiang Guo.

**Data curation:** Shuang Deng, Zhongqiang Guo.

**Formal analysis:** Shuang Deng, Zhongqiang Guo, Siyuan He.

**Investigation:** Shuang Deng, Zhongqiang Guo.

**Methodology:** Shuang Deng, Zhongqiang Guo, Siyuan He.

**Project administration:** Shuang Deng, Zhongqiang Guo, Siyuan He, Xuetao Sun.

**Resources:** Shuang Deng, Zhongqiang Guo, Siyuan He.

**Software:** Shuang Deng, Zhongqiang Guo, Siyuan He, Xuetao Sun.

**Supervision:** Shuang Deng, Zhongqiang Guo.

**Validation:** Shuang Deng, Zhongqiang Guo, Siyuan He, Xuetao Sun.

**Visualization:** Shuang Deng, Zhongqiang Guo, Xuetao Sun.

**Writing – original draft:** Shuang Deng.

**Writing – review & editing:** Zhongqiang Guo.

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
