## [Decision Letter · Decision Letter 0]

29 Aug 2025

Dear Dr. Guo,

Thank you for submitting your manuscript to PLOS ONE. After careful consideration, we feel that it has merit but does not fully meet PLOS ONE’s publication criteria as it currently stands. Therefore, we invite you to submit a revised version of the manuscript that addresses the points raised during the review process.

**ACADEMIC EDITOR: Major revision**

We look forward to receiving your revised manuscript.

Kind regards,

Marwan Salih Al-Nimer, MD, PhD

Academic Editor

PLOS ONE

2. Please note that your Data Availability Statement is currently missing [the repository name and/or the DOI/accession number of each dataset OR a direct link to access each database]. If your manuscript is accepted for publication, you will be asked to provide these details on a very short timeline. We therefore suggest that you provide this information now, though we will not hold up the peer review process if you are unable.

Additional Editor Comments:

This article is not well prepared. I suggest to take considerations to the:

1: Title

2: keywords

3:Abbreviations

4: Scoring, e.g., Likert 4

5: phases of CKM

6: figures and Tables are not clear and tables missed footnotes

7: statistical analysis: rephrase to be clear

8: discussion is weak

Reviewers' comments:

Reviewer's Responses to Questions

**Comments to the Author**

1. Is the manuscript technically sound, and do the data support the conclusions?

Reviewer #1: Yes

2. Has the statistical analysis been performed appropriately and rigorously?

Reviewer #1: Yes

3. Have the authors made all data underlying the findings in their manuscript fully available?

Reviewer #1: Yes

4. Is the manuscript presented in an intelligible fashion and written in standard English?

Reviewer #1: Yes

Reviewer #1: The manuscript addresses an important and timely topic: the interplay between social isolation, cardiometabolic burden, and physical dysfunction in aging Chinese adults, using nationally representative CHARLS data. The integration of social and biological determinants, threshold analyses, and causal mediation models is innovative and of interest to the aging and public health community. The findings have potential policy implications for integrated interventions targeting both social and biomedical risk factors.

However, several issues need to be addressed before the manuscript can be considered for publication. These include clarification of methodology, further justification of analytical choices, and improvements in presentation, interpretation, and discussion of limitations.

1. Study Design and Causality

- The manuscript acknowledges the cross-sectional nature of the study but still makes causal inferences (e.g., “EPWV → frailty → physical dysfunction” as a pathway). The language in the results and discussion should be moderated to avoid implying causality without longitudinal validation.

- Consider clarifying that mediation analyses in cross-sectional data can only suggest potential pathways and require prospective confirmation.

2.Operationalization of Social Isolation

- The social isolation score derivation requires more detail. Please specify which CHARLS items were used, their weighting, and whether the measure has been validated in the Chinese population.

- Indicate how missing data in social isolation components were handled, as imputation could affect classification.

3.Biomarker Threshold Determination

- The approach for identifying inflection points (threshold effect analysis) should be described more explicitly. Were spline regressions, piecewise linear models, or other non-linear models used? How was the optimal cut-point determined and validated?

- Consider presenting sensitivity analyses to show robustness of thresholds.

4. Confounding and Effect Modification

- While demographic, lifestyle, and socioeconomic covariates were adjusted for, residual confounding is likely. Variables such as depressive symptoms, physical activity, and nutritional status may influence both social isolation and physical function.

- Some effect modification findings (e.g., stronger effects in males <60 years) warrant deeper exploration — could occupational patterns, migration history, or gender norms explain this?

5. Interpretation of Mediation Results

The reported mediation effect of frailty (57.8%) is large. Please clarify whether the frailty index shares overlapping components with the physical dysfunction outcome, as this could inflate mediation estimates.

Discuss possible bi-directionality between frailty and vascular aging in more depth.

6.Generalizability

- As the data are from 2015, please discuss whether changes in social structures, healthcare access, or biomarker profiles in recent years might limit applicability to the current Chinese older adult population.

Recommendations for the Authors

- Temper causal language and clearly state the observational nature of the study.

- Provide greater detail on construction and validation of the social isolation index.

- Explain threshold modeling methods and include sensitivity checks.

- Address potential measurement overlap between frailty and physical function.

- Expand on contextual factors that could explain subgroup differences.

- Streamline tables/figures for clarity and reader engagement.

**Do you want your identity to be public for this peer review?** For information about this choice, including consent withdrawal, please see our Privacy Policy

Reviewer #1: No

---

## [Author Response · Author response to Decision Letter 1]

6 Oct 2025

Point-by-point Response Letter-[Manuscript ID:PONE-D-25-39258]

Title:Social Isolation and Cardiometabolic Burden Synergistically Predict Physical Dysfunction in Aging Chinese Adults: Evidence of Risk Thresholds and the Mediating Role of Frailty

Dear Editors and Reviewers of Plos One,

We are deeply grateful for considering our manuscript for Plos One and value the insightful comments offered. These suggestions have been crucial for enhancing our manuscript's quality, and we're confident the revisions have significantly improved its clarity, rigor, and impact. We've meticulously revised the manuscript, addressing each comment with careful consideration and research. Besides responding to the feedback, we've also thoroughly reviewed the entire manuscript, making additional refinements to ensure top - notch scientific and literary quality. All changes are clearly tracked in the revised manuscript, and a clean version is also prepared. Below are our detailed point-by-point responses to each comment. We've aimed to be comprehensive and transparent in our explanations, providing evidence and reasoning for the changes made. Thank you again sincerely for your dedication and hard work. Your expertise is invaluable to the scientific community, and we're honored to contribute to this prestigious journal.

Yours Sincerely,

ZhongQiang Guo

Henan University

guozhongqiang0701@gmail.com

Editor

Comments/suggestions:

1:Title

Response:Thank you for the valuable feedback from the editor,The updated title highlights the core variables (Social Isolation, CKM, Physical Dysfunction, Frailty) and main findings (Mediating Role, Risk Thresholds)

Revised version: Social isolation, cardiometabolic syndrome, and physical dysfunction in aging chinese adults: the pivotal mediating role of frailty and evidence of risk thresholds

2:keywords

Response:Thank you for the valuable feedback from the editor,Deleted the typo "pulse circulation disorders" and replaced it with the overly broad "lipid metabolism disorders"

Revised version: Modified keywords:social isolation;cardiovascular-kidney-metabolic (CKM) syndrome;frailty;physical dysfunction;threshold effect;CHARLS.

3: Abbreviations

Response:We sincerely thank the editor for this important comment. We have thoroughly reviewed and standardized the use of abbreviations throughout the manuscript to ensure clarity and consistency for readers. The specific measures we have taken are as follows:

First Use Definition: Every abbreviation is now explicitly defined upon its first appearance in the main text, including the Abstract, Introduction, Methods, Results, and Discussion. The format used is "Full Term (Abbreviation)".

Example from the Abstract: "...cardiovascular–kidney–metabolic (CKM) syndrome... estimated pulse wave velocity [EPWV]..."

Creation of a Standalone Abbreviations Section: We have added a dedicated "Abbreviations" section in the manuscript, located after the "Acknowledgments" section and before the "References" list. This section provides an alphabetical list of all major abbreviations used in the paper for quick reference.

Consistency Check: We have ensured that each abbreviation is used consistently after its initial definition and that no undefined abbreviations appear in the text.

We believe these revisions have significantly improved the readability of the manuscript and prevented any potential confusion regarding the terminology used.

Revised version: PD, Physical Dysfunction; SI, Social Isolation; EPWV, Estimated Pulse Wave Velocity; ASM, Accessory Skeletal Muscle Mass; VAI, Visceral Adiposity Index; eGFR, Estimated Glomerular Filtration Rate; CTI, C-reactive protein triglyceride glucose index.

4: Scoring, e.g., Likert 4

Response:We thank the editor for raising this crucial point regarding the clarity of our scoring methods. We have comprehensively revised the manuscript to provide explicit and detailed descriptions of how all key composite variables were constructed and scored. The specific clarifications are as follows:

Revised version: (lines 170-269) in the “Materials and methods” section.

Outcome variable

Physical dysfunction

Calculation formula and assignment rules: The evaluation items include 9 basic physical function activities: running or jogging for 1 kilometre; walking 1 kilometre; walking 100 metres; rising after sitting for a long time; continuously climbing several layers of stairs; bending or squatting; raising arms over shoulders; picking up small coins from a table; and carrying heavy objects (>10 pounds). Answer options and coding: 0 for no difficulty, 1 for difficulty but still able to complete, 2 for difficulty and help required, and 3 for inability to complete. If a subject reported difficulty (coding ≥ 1) in any of the 9 items, this was defined as physical dysfunction [24].

Core independent variables

Social Isolation (SI): The construction in CHARLS research is based on five core items: marital status (1 point for unmarried, divorced, or widowed), living arrangement (1 point for living alone), social contact frequency with children (1 point for less than once a week), type of residence (1 point for rural areas), and participation in social activities (1 point for not participating in any social activities during the past month).

If each item met the conditions, it was given a score of 1 point. The total score ranges from 0 to 5 points. The higher the score is, the more serious the social isolation.

Finally, a total score of ≥ 2 points is considered as the critical point for judging social isolation (i.e., "yes"), and a total score of <2 points is judged as non-social isolation ("no").

This scoring standard refers to the international health and retirement research (HRS) model, and each item is simply summed with equal weight (i.e., 1 point for each item) [25].

This critical value (≥ 2 points) has been verified in many international studies to effectively predict health outcomes such as mortality, cardiovascular disease, and chronic kidney disease. For example, a meta-analysis [26,27] revealed that social isolation increased the risk of all-cause death by 29% (OR=1.29), coronary heart disease by 29%, and stroke by 32%.

In the Chinese population, the CHARLS cohort study (the sample covers middle-aged and elderly people in 28 provinces across the country) further confirmed [25] that the risk of renal function decline in social isolation with the same critical value (≥ 2 points) significantly increased, indicating that the standard has predictive validity for the Chinese population.

With respect to the processing of missing data, CHARLS routinely uses statistical techniques to fill in missing values (such as mode for classified variables and mean for continuous variables) and corrects sampling deviation by analysing weights to ensure national representativeness [28,29].

However, in the specific operation of the social isolation component, if the data of five core items are missing, this study chooses to directly eliminate the sample rather than to complete it.

This approach may reduce the sample size but avoid the classification bias that may be introduced by imputation, such as the lack of marital status or social activity participation. If the mode is filled, some actual isolators may be erroneously classified into a nonisolated group [25].

Of note [30], mentioned that other studies used different threshold values (such as ≥ 4 points to determine isolation), but CHARLS mainly used the international standard of ≥ 2 points and noted that the reliability and validity of this standard in the elderly population has been supported by many studies [31].

Cardiovascular–kidney–metabolic (CKM) syndrome stages

Stage 0: no CKM risk factors (no overweight/obesity, metabolic abnormalities, CKD, or cardiovascular disease).

Stage 1: overweight/obesity (BMI ≥ 25 kg/m²) or abdominal obesity but no other metabolic risk factors (such as hypertension or diabetes) or CKD.

Stage 2: presence of metabolic risk factors (hypertension, hypertriglyceridaemia, metabolic syndrome, diabetes) or CKD (eGFR<60 mL/min/1.73 m² or proteinuria) but no cardiovascular disease.

Stage 3: at least one case of subclinical cardiovascular disease (such as atherosclerosis or subclinical heart failure) accompanied by overweight/obesity, metabolic risk factors, or CKD.

Stage 4: clinically diagnosed cardiovascular diseases (such as coronary heart disease, heart failure, and stroke) with at least one other risk factor (CKD, BMI ≥ 25 kg/m², diabetes, hypertension, etc.) are divided into no renal failure (4a) and renal failure (4b, eGFR<15 mL/min/1.73 m² or renal replacement therapy) [32,33].

Vascular ageing biomarkers

Estimated pulse wave velocity (EPWV)

EPWV is calculated on the basis of the MBP and age using the following formula: EPWVageageageMBP) age MBP MBP), where [34].

Body Composition Indicators:

Accessory skeletal muscle mass (ASM): Male sex=1, female sex=2, sum of skeletal muscle mass in arms and legs (kg), and assessment of muscle mass [35].

Frailty Index=, which is a comprehensive assessment of individual physiological reserve decline and vulnerability [36].

Frailty: Using a Fried phenotypic model, weakness was defined as a score ≥ 3 for the following five criteria: involuntary weight loss (weight loss ≥ 4.5 kg or 5% in the past year), self-reported fatigue (CES-D assessment), a decrease in grip strength (BMI and sex correction), slow walking speed (4-metre walking test), and low physical activity (IPAQ questionnaire assessment) [37].

When scoring, each criterion is a binary variable (existence=1 point; nonexistence=0 points). If the total score is ≥ 3 points, it is judged as frailty, and if the total score is <3 points, it is judged as non-frailty.

(yes/no) [38].

Visceral Adiposity Index (VAI):

Indicators reflecting visceral fat function and quantity were calculated on the basis of waist circumference, body mass index, triglyceride levels, and high-density lipoprotein cholesterol levels [39].

Core indicators of renal function

Estimated Glomerular Filtration Rate, eGFR:

Female:

Male:

followed by the assessment of renal function [40].

Lipid metabolism indicators

C-reactive protein triglyceride glucose index (CTI): The formula for detecting C-reactive protein, triglyceride, and glucose levels in blood samples is as follows:

As a novel composite biomarker that integrates inflammation and lipid metabolism, CTI has shown significant potential in predicting neurodegenerative diseases [42].

References

24. Resnik L, Gozalo P, Hart DL. Weighted index explained more variance in physical function than an additively scored functional comorbidity scale. J Clin Epidemiol. 2011;64: 320-330.

25. Zhou W, Li Y, Ning Y, Gong S, Song N, Zhu B, et al. Social isolation is associated with rapid kidney function decline and the development of chronic kidney diseases in middle-aged and elderly adults: findings from the China health and retirement longitudinal study (CHARLS). Front Med (Lausanne). 2021;8: 782624.

26. Valtorta NK, Kanaan M, Gilbody S, Ronzi S, Hanratty B. Loneliness and social isolation as risk factors for coronary heart disease and stroke: systematic review and meta-analysis of longitudinal observational studies. Heart. 2016;102: 1009-1016.

27. Holt-Lunstad J, Smith TB, Baker M, Harris T, Stephenson D. Loneliness and social isolation as risk factors for mortality: a meta-analytic review. Perspect Psychol Sci. 2015;10: 227-237.

28. Zhao Y, Hu Y, Smith JP, Strauss J, Yang G. Cohort profile: the China health and retirement longitudinal study (CHARLS). Int J Epidemiol. 2014;43: 61-68.

29. Song Y, Zhu C, Shi B, Song C, Cui K, Chang Z, et al. Social isolation, loneliness, and incident type 2 diabetes mellitus: results from two large prospective cohorts in Europe and East Asia and Mendelian randomization. EClinicalMedicine. 2023;64: 102236.

30. Qin W, Xiang X, Taylor H. Driving cessation and social isolation in older adults. J Aging Health. 2020;32: 962-971.

31. Lyu C, Siu K, Xu I, Osman I, Zhong J. Social isolation changes and long-term outcomes among older adults. JAMA Netw Open. 2024;7: e2424519.

32. Roa BPA, Varne SRR. Evaluation of cardiovascular kidney metabolic syndrome and cardiovascular hepato-renal metabolic syndrome (CHARM): from encryption to decryption. Int J Clin Exp Physiol. 2024;10: 98-102.

33. Theodorakis N, Nikolaou M. From cardiovascular-kidney-metabolic syndrome to cardiovascular-renal-hepatic-metabolic syndrome: proposing an expanded framework. Biomolecules. 2025;15: 213.

34. Li D, Cao F, Cheng W, Xu Y, Yang C. Predictive value of estimated pulse wave velocity for cardiovascular and all-cause mortality in individuals with obesity. Diabetol Metab Syndr. 2023;15: 40.

35. Ji W, Liu X, Zhang Y, Zhao Y, He Y, Cui J, et al. Development of formulas for calculating L3 skeletal muscle mass index and visceral fat area based on anthropometric parameters. Front Nutr. 2022;9: 910771.

36. Rockwood K, Nassar B, Mitnitski A. Apolipoprotein E-polymorphism, frailty and mortality in older adults. J Cell Mol Med. 2008;12: 2754-2761.

37. Bautista MAC, Malhotra R. Identification and measurement of frailty: a scoping review of published research from Singapore. Ann Acad Med Singap. 2018;47: 455-491.

38. Quach J, Kehler DS, Giacomantonio N, McArthur C, Blanchard C, Firth W, et al. Association of admission frailty and frailty changes during cardiac rehabilitation with 5-year outcomes. Eur J Prev Cardiol. 2023;30: 807-819.

39. Kaminska MS, Lubkowska A, Panczyk M, Walaszek I, Grochans S, Grochans E, et al. Relationships of body mass index, relative fat mass index, and waist circumference with serum concentrations of parameters of chronic inflammation. Nutrients. 2023;15: 2789.

40. Yu S, Yang H, Guo X, Zheng L, Sun Y. Association between obese phenotype and mildly reduced eGFR among the general population from rural Northeast China. Int J Environ Res Public Health. 2016;13: 540.

41. Ren Y, Xu R, Zhang J, Jin Y, Zhang D, Wang Y, et al. Association between the C-reactive protein-triglyceride-glucose index and endometriosis: a cross-sectional study using data from the national health and nutrition examination survey, 1996-2006. BMC Womens Health. 2025;25: 13.

42. Li Q, Song Y, Zhang Z, Xu J, Liu Z, Tang X, et al. The combined effect of triglyceride-glucose index and high-sensitivity C-reactive protein on cardiovascular outcomes in patients with chronic coronary syndrome: a multicenter cohort study. J Diabetes. 2024;16: e13589.

5: phases of CKM

Response:

We thank the editor for this critical comment. We acknowledge that the original manuscript lacked a detailed definition of the cardiovascular-kidney-metabolic (CKM) syndrome staging, which is essential for understanding the exposure gradient. In response, we have now provided a comprehensive and clear description of the CKM staging criteria in the revised manuscript.

Specifically, in the "Variable Definition and Measurement" section (Section 2.2.2, under "Core independent variables"), we have added a new subsection titled "Cardiovascular–kidney–metabolic (CKM) syndrome stages". This subsection details the operational definitions for all five stages (0 to 4), based on the seminal framework proposed by the American Heart Association (AHA) and subsequent operationalizations for epidemiological studies [32, 33].

Revised version: (lines 217-230) in the “Materials and methods” section.

Cardiovascular–kidney–metabolic (CKM) syndrome stages

Stage 0: no CKM risk factors (no overweight/obesity, metabolic abnormalities, CKD, or cardiovascular disease).

Stage 1: overweight/obesity (BMI ≥ 25 kg/m²) or abdominal obesity but no other metabolic risk factors (such as hypertension or diabetes) or CKD.

Stage 2: presence of metabolic risk factors (hypertension, hypertriglyceridaemia, metabolic syndrome, diabetes) or CKD (eGFR<60 mL/min/1.73 m² or proteinuria) but no cardiovascular disease.

Stage 3: at least one case of subclinical cardiovascular disease (such as atherosclerosis or subclinical heart failure) accompanied by overweight/obesity, metabolic risk factors, or CKD.

Stage 4: clinically diagnosed cardiovascular diseases (such as coronary heart disease, heart failure, and stroke) with at least one other ri

---

## [Editor Report · Decision Letter 1]

13 Oct 2025

Social Isolation and Cardiometabolic Burden Synergistically Predict Physical Dysfunction in Aging Chinese Adults: Evidence of Risk Thresholds and the Mediating Role of Frailty

PONE-D-25-39258R1

Dear Dr. Zhongqing Guo,

We’re pleased to inform you that your manuscript has been judged scientifically suitable for publication and will be formally accepted for publication once it meets all outstanding technical requirements.

Kind regards,

Marwan Salih Al-Nimer, MD, PhD

Academic Editor

PLOS ONE

Additional Editor Comments (optional):

No comment
---

## [Editor Report · Acceptance letter]

PONE-D-25-39258R1

PLOS ONE

Dear Dr. Guo,

I'm pleased to inform you that your manuscript has been deemed suitable for publication in PLOS ONE. Congratulations! Your manuscript is now being handed over to our production team.

Kind regards,

on behalf of

Professor Marwan Salih Al-Nimer

Academic Editor

PLOS ONE